# Persuasive COVID-19 vaccination campaigns on Facebook and nationwide vaccination coverage in Ukraine, India, and Pakistan

**Maike Winters**[1,2]*, **Sarah Christie**[1,2], **Chelsey Lepage**[3], **Amyn A. Malik**[1,2],
**Scott Bokemper**[4,5], **Surangani Abeyesekera**[6], **Brian Boye**[7], **Midhat Moini**[7], **Zara Jamil**[8],
**Taha Tariq**[8], **Tamara Beresh**[9], **Ganna Kazymyrova**[9], **Liudmyla Palamar**[9], **Elliott Paintsil**[1],
**Alexandra Faller**[10], **Andreea Seusan**[10], **Erika Bonnevie**[10], **Joe Smyser**[10],
**Kadeem Khan**[11], **Mohamed Gulaid**[11], **Sarah Francis**[12], **Joshua L. Warren**[13],
**Angus Thomson**[3], **Saad B. Omer**[1,2,13]

**1** Yale Institute for Global Health, Yale University, New Haven, Connecticut, United States of America, **2** Yale School of Medicine, Yale University, New Haven, Connecticut, United States of America, **3** Irimi Company, Lyon, France, **4** Institution for Social and Policy Studies, Yale University, New Haven, Connecticut, United States of America, **5** Center for the Study of American Politics, Yale University, New Haven, Connecticut, United States of America, **6** UNICEF Headquarters, New York, NY, United States of America, **7** UNICEF Country Office India, New Delhi, India, **8** UNICEF Country Office, Karachi, Pakistan, **9** UNICEF Country Office, Kyiv, Ukraine, **10** The Public Good Projects, Alexandria, Virginia, United States of America, **11** Meta Platforms Inc., Menlo Park, California, United States of America, **12** Team Upswell, Seattle, Washington, United States of America, **13** Peter O'Donnell Jr. School of Public Health, University of Texas Southwestern, Dallas, Texas, United States of America

\* maike.winters@yale.edu

## Abstract

Social media platforms have a wide and influential reach, and as such provide an opportunity to increase vaccine uptake. To date, there is no large-scale, robust evidence on the offline effects of online messaging campaigns. We aimed to test whether pre-tested, persuasive messaging campaigns from UNICEF, disseminated on Facebook, influenced COVID-19 vaccine uptake in Ukraine, India, and Pakistan. In Ukraine, we deployed a stepped-wedge randomized controlled trial (RCT). Half of the 24 oblasts (provinces) received five weeks of the intervention, the other half ten weeks of the intervention. In India, an RCT with an augmented synthetic control was conducted in five states (Bihar, Chhattisgarh, Jharkhand, Madhya Pradesh, Rajasthan), whereby 40 out of 174 districts were randomized to receive six weeks of intervention. In Pakistan we deployed a pre-post design, whereby 25 city districts received six weeks of the intervention. Weekly COVID-19 vaccination data was sourced through government databases. Using Poisson regression models, the association between the intervention and vaccine uptake was estimated. In Ukraine we conducted a survey among Facebook users at three time points during the RCT, to ascertain vaccination intentions and trust in vaccines. The campaigns reached more than 110 million Facebook users and garnered 2.9 million clicks. In Ukraine, we found that the intervention did not affect oblast-level vaccination coverage (Relative Risk (RR): 0.93, 95% Confidence Interval (CI) 0.86–1.01). Similarly, in India and Pakistan we found no effect of our intervention (India: RR 0.85, 95% CI 0.70–1.04; Pakistan: RR 0.64, 95% CI 0.01–29.9). The survey among Facebook users in Ukraine showed that trust in vaccines and information

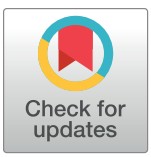

**Data Availability Statement:** The surveys we conducted in Ukraine, as well as COVID-19 weekly vaccination from Ukraine and India are available at

OSF: https://osf.io/27t6r/ COVID-19 vaccination data from Pakistan was obtained through a confidential agreement with Pakistan's National Command and Operation Center and cannot be shared publicly.

**Funding:** This study was funded by Facebook Inc. (now Meta Platforms Inc.), grant number: GR112635, received by S.B.O, supporting the time commitment of M.W., S.C., A.A.M, S.B., E.P., J.W, S.B.O. C.L, K.K., M.G. and S.F. received salaries from Meta during (parts of) the study. Meta's website: https://about.meta.com/ J.W. was also supported through the NIAID (grant number R01 AI37093). The funders had no role in study design, data collection and analysis or decision to public. The co-authors who received salaries from Meta reviewed the manuscript.

**Competing interests:** I have read the journal's policy and the authors of this manuscript have the following competing interests: Funding for this work was received from Facebook Inc. (now Meta Platforms Inc.). Four co-authors (CL, KK, MG, SF) were during the study (or parts of the study) salaried Meta employees.

sources was an important predictor of vaccination status and intention to get vaccinated. Our campaigns on Facebook had a wide reach, which did not translate in shifting behaviours. Timing and external events may have limited the effectiveness of our interventions.

## Introduction

Since the COVID-19 vaccines became available to the public, there has been an increased focus on vaccine hesitancy and acceptance. Vaccine decision-making is a complex and context-specific behavior which may be variously mediated by attitudes, trust, social norms, moral values, beliefs, worldview as well as structural barriers [1–3]. Vaccine acceptance lies along a continuum; on one side acceptance of the recommended vaccines and on the other end refusal of all vaccines. In between are those who might be hesitant towards some vaccines but could be amenable to changing their attitudes [4].

The 'infodemic' has caused an overload of (mis)information, making it difficult for the public to discern between trustworthy information and misinformation [5]. Viral misinformation has not only undermined efforts of public health agencies and eroded trust, but also potentially increased vaccine hesitancy [6, 7]. Social media platforms, with their global reach, can be a predominant source of vaccine misinformation and disinformation [8, 9].

At the same time, the reach and influence of social media could potentially be harnessed in a positive way; to build vaccine knowledge and confidence, promote demand, and ultimately increase vaccine uptake [8, 9]. Strategic campaign marketing tools available online such as audience insights, segmentation, and A/B message testing capability that drive consumer behavior toward goods and services can be adapted to health communications for public health impact [10, 11]. Messages that are evidence-based, context-driven, and grounded in behavioral science have greater potential to influence attitudes towards vaccines, public trust, and health behavior, than generic, one-size-fits-all messages, but research is limited in converting these insights into effective digital messaging campaigns in low-and middle-income countries [12, 13].

Studies that have investigated social media campaigns to promote vaccine attitudes and uptake have provided mixed results to date [14–16]. A randomized controlled trial (RCT) in Israel targeting mothers to promote HPV vaccination among their teenage daughters found a slight increase in vaccine uptake in daughters from medium-to-low socio-economic backgrounds, while it decreased uptake in lower socio-economic groups [14]. A Facebook campaign in Philadelphia targeting adolescents for HPV vaccination reached more than 155,000 people, but only two teenagers got vaccinated because of the campaign [15]. Another study found a moderate increase in HPV vaccine awareness following a social media campaign, but no change in participants' intention to get vaccinated [16]. Evidence suggests that vaccine messaging may often be ineffective, and in some instances may even backfire, decreasing intention to vaccinate, in particular in people who were already hesitant [17, 18]. The framing of messages and the trustworthiness of the messengers play an important role in vaccine acceptance, yet promotion messages are often not tested for efficacy and safety before dissemination [12, 14, 16]. A major current research gap is the lack of large, rigorously designed studies to more accurately determine the effectiveness of context-driven vaccine promotion messages disseminated through social media in promoting vaccine uptake [19, 20].

While studies to date have investigated the effectiveness of social media interventions on influencing vaccine attitudes and vaccination intentions in highly controlled settings [20–24],

there is a research gap in large-scale robust studies that investigate the real-world effect of social media messaging on vaccination coverage. This large study aimed to assess whether surfacing behavioral insights, designing communications for local contexts, and testing and refining these interventions *online* will increase vaccine acceptance and vaccine uptake *offline*. The primary goal of this work was to understand whether different types of persuasive messaging disseminated on Facebook, in partnership with the United Nations Children's Fund (UNICEF), can influence regional COVID-19 vaccination coverage in Ukraine, India, and Pakistan. Secondly, in Ukraine we assessed whether these messages influenced self-reported vaccination uptake and intention to get vaccinated, as well as attitudes towards vaccines.

## Methods

### Study designs and randomization

For all three countries the feasibility to conduct an RCT was assessed. However, due to the availability and granularity of COVID-19 vaccination data and the varying geographies, three different prospective study designs were employed for Ukraine, India, and Pakistan.

**Ukraine.** In Ukraine, a stepped-wedge RCT was designed, with the 24 'oblasts' (i.e., provinces, Kyiv region and Kyiv city were combined into one oblast) of the country as clusters. While the number of clusters was relatively small, the stepped-wedge design enhanced the statistical power of the trial. With this design, in Step 0 (the first five weeks) no oblasts received intervention. At Step 1, in the second five-week period, half of the oblasts received intervention, while the other half did not receive intervention. At Step 2 all oblasts received intervention for the final five weeks, see S1 Fig and S1 Table for an overview of the study design. To determine which oblasts would receive the intervention for five weeks, we conducted a stratified randomization that balanced the intervention and control groups on observed COVID-19 vaccination coverage (from the week before the start of the study), population, and the most recently available estimates of the percentage of the population who lived in an urban setting, see S2 Table. The study was powered to detect a difference between proportions of 0.00056 with 80% power. The study took place between November 8, 2021 and February 24, 2022.

A survey, targeted at Facebook users in Ukraine, was conducted at three time points during the first, second, and third step of the RCT. Recruitment ads for the survey were on Facebook for one week during these periods. The survey was hosted on Qualtrics and contained questions on demographics, self-reported COVID-19 vaccination, attitudes towards vaccination, perceived safety of the vaccines, trust in the information around COVID-19 vaccines from key actors in the pandemic (such as government, Ministry of Health, doctors, media, and family/friends), and the Vaccine Trust Indicator, a scale of six items [25]—the full survey can be found in S1 Text. A fourth survey period was planned to be conducted a few weeks after the end of the intervention, but due to the Russian invasion in Ukraine, this was no longer feasible.

**India.** In India, the study was deployed in five states (Bihar, Chhattisgarh, Jharkhand, Madhya Pradesh, and Rajasthan). Due to the large number of districts in these states (174), we were able to use a parallel RCT, whereby 40 of the 174 districts were randomized to receive the intervention messages for six weeks, see S3 Table. We used an augmented synthetic control method [26, 27], utilizing the GeoLift tool from Facebook to run simulations to identify a combination of districts that would produce a well-powered study. For this, we used data from the first 53 weeks of the COVID-19 vaccination in the five states, from January 16, 2021 to January 15, 2022. The simulations determined that the number of districts that should be assigned to intervention should be 40, to have 80% power to detect an increase in vaccination of 2%. Districts that were assigned to intervention received the posts on Facebook between February 19,

2022 and March 31, 2022. The intervention posts were optimized for reach so that the Facebook algorithm delivered the posts to as many users as possible.

**Pakistan.** In Pakistan, due to limits in the availability of COVID-19 vaccination data, low statistical power and a preference of the stakeholders to message in all selected city districts, we opted for a pre-post design, whereby we followed 25 city districts during a period before the intervention, followed by a period of six weeks when all city districts received the intervention. The 25 city districts were prioritized by the Ministry of National Health Services, Regulations and Coordination for intervention due to suboptimal coverage, see S4 Table. Our power analysis showed that with 80% power, we could detect a difference of 0.031 between the vaccine coverage in the treatment and control periods. Like the campaigns in Ukraine and India, the posts were optimized for reach on Facebook. The intervention took place between February 5, 2022 and March 18, 2022.

The studies were funded by Meta and approved by the Yale Institutional Review Board (protocol number: 2000031351). Informed consent was obtained before the start of the surveys in Ukraine. Potential participants were asked to read our information sheet online, containing information about the study as well as contact information for the Yale research team. Participants were then asked for their consent. Those who consented continued to the main survey. The study protocol and analysis plan for each country was registered with Open Science Framework prior to data collection: https://osf.io/27t6r/

### Intervention

The intervention comprised context-driven, evidence-based messages on Facebook that target audiences living in randomized geographical areas. The control condition received no messages. In Ukraine, the treatment status of each region changed over time, with all regions beginning in the control condition.

The intervention was co-designed with UNICEF Headquarters and Country Offices in India, Pakistan, and Ukraine. UNICEF is a global organization that seeks to improve the health and well-being of children worldwide and served during the pandemic as the lead supply chain provider for COVAX [28] In each country, 5–8 ads were selected for the study and featured the UNICEF logo, see Fig 1. All ads contained a link to a website with more information on COVID-19 vaccination and a vaccination portal.

Messaging campaigns were designed based on a detailed landscape analysis that included integration of data and insights from multiple sources: public posts analysis on Facebook (developed by Meta's Data for Good team), routine survey data from UNICEF, as well as a review of the peer-reviewed literature. Using the WHO's and UNICEF's Behavioral and Social Drivers (BeSD) of Vaccination Framework [29, 30], levers were identified that related to 'thinking and feeling' (e.g., perceived disease risk), 'social processes' (e.g., social norms), 'motivation' (e.g., willingness to get vaccinated) and 'practical issues' (e.g., accessibility of vaccines).

These messages were then iteratively tested using Facebook's Brand Lift Surveys (BLS); an A/B testing method whereby Facebook users were randomly exposed to either the message or not [31]. Such methods are typically used by corporations to test ads for brand awareness and their potential to drive consumer behavior to goods, products and services [32]. A five-item survey that assessed the recall of the message, perceived importance of COVID-19 vaccines, whether individuals would recommend vaccines to friends/family and other relevant constructs based on ad content was administered to understand their potential to lift attitudes.

Results from the BLS showed which campaigns were most effective at changing attitudes towards vaccines and which advertisements within the campaigns were highest performing in terms of user engagement, recall, and clickthrough rate (defined as the proportion of

Ukraine India Pakistan

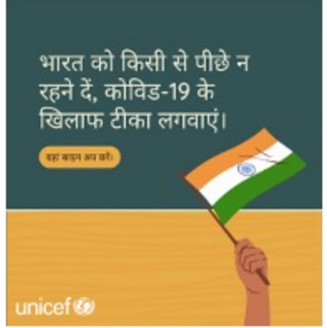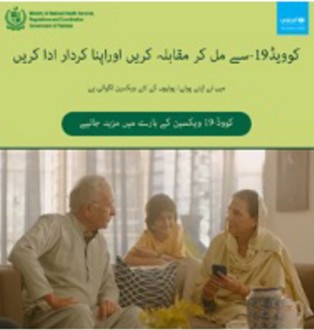

On Asset: Your help is necessary to protect others from COVID-19.
Help your loved ones find their nearest vaccination point or center.
Learn More

Accompanying text: You are doing a really great thing when you help people get vaccinated against COVID-19. Learn more at 0 800 60 20 19 or vaccination.COVID-19.gov.ua

On Asset: Don't let India down, get vaccinated against COVID-19.
Sign up here

Accompanying text: We can win the fight against COVID-19 if everyone gets vaccinated. We have sacrificed too much to let India down now.
#TogetherAgainstCOVID19
#EndCOVID19 #VaccinesWork
Sign Up:
https://www.cowin.gov.in/home

On Asset: Do your part. Together against COVID-19. We got vaccinated for our grandchildren.
Learn More about COVID-19 vaccines

Accompanying text: It was hard staying apart from my family, but now that I'm vaccinated we can stay safe together. Protect yourself by getting the COVID-19 vaccine.
#TogetherAgainstCOVID19
#VaccinesForPakistan"

**Fig 1. Example of ads in each country.**

individuals who clicked on the ad out of the total number of times users were exposed to the ad). Results from this testing informed the final set of messaging interventions that were used for this study. The final set of messages can be found in the online repository: https://osf.io/27t6r/. Briefly, the ad set in Ukraine comprised ads with practical information, safety and efficacy information, liberty-framed posts, and a social responsibility post. In India, the ad set tapped into national pride, safety of the vaccines and social responsibility. In Pakistan, the ads showed social responsibility and testimonials with vaccine accessibility information. All ads contained a link to a website with more information on COVID-19 vaccination and a vaccination scheduler.

Ads in India and Ukraine featured illustrations that were converted in GIFs to enhance user engagement; ads in Pakistan were framed as photographic testimonials, and co-branded with UNICEF logo and Ministry of Health services. In each context, ads used a mix of content that was informed by Moral Foundations Theory, which emphasized values-based messaging to drive behavior change (in Ukraine, ads focused on a liberty-frame; in Pakistan, ads focused on an equity frame, and in India, ads focused on national pride) [1, 33, 34]. Furthermore, ads featured trusted messengers in such contexts, which were identified in the brand lift study, although these differed by context (I.e., doctors and scientists were featured in India and Ukraine, alongside relatable family characters in Ukraine and Pakistan, to instill social responsibility for COVID-19 vaccines). Ads also emphasized safety and efficacy of vaccines and provided basic information on how to access vaccines.

The UNICEF Country Offices in Ukraine, India, and Pakistan led the digital intervention campaigns. Ad credits for the campaigns were provided by Meta. The campaigns were run in Ukrainian in Ukraine, in Hindi, English and Urdu in India, and in English and Urdu in Pakistan. The ad campaigns ran on Facebook and were optimized for reach, to ensure maximum exposure to Facebook users 18–65+ years of age.

## Participants

The final set of ads for each country was posted in the oblasts (Ukraine) and districts (India) that were randomized to the intervention, using Facebook's Ads Manager to create the geographical borders. In Pakistan, Facebook users in city districts with radii to encompass the entire district were targeted during the intervention period. In all countries, Facebook users aged 18 and older were targeted, reflecting the population eligible at that time for COVID-19 vaccines. The Facebook user base in Ukraine at the time of the study was between 22.3–24.8 million users, roughly 60% of the adult population. In India, there were approximately 81.9 million Facebook users in the 40 intervention districts, which corresponded to about 36% of the adult population in those states. Lastly, in Pakistan, the Facebook user base in the 25 city districts was around 32.9 million users, or around 52% of those eligible for COVID-19 vaccines in those districts.

Recruitment of the survey that was conducted in Ukraine was similarly done via Facebook, using ads featuring the UNICEF logo, inviting people (Facebook users 18 years and older) to share their thoughts on COVID-19 vaccines, with a link to the survey.

## Outcomes & statistical analysis

**Primary outcomes.** The primary outcome of these studies was the regional COVID-19 vaccination coverage across time. This was measured using the weekly updates on the number of first and second doses by oblast (Ukraine), district (India), and city/district (Pakistan). In Ukraine, weekly COVID-19 vaccination data were obtained through a database maintained by the Ukrainian government (https://health-security.rnbo.gove.ua/). In India, weekly vaccination doses were ascertained through the COWIN Dashboard, which was maintained by the Ministry of Health in India. UNICEF Pakistan provided weekly COVID-19 vaccination data, sourced from a database maintained by the Government of Pakistan.

To understand whether the intervention had an effect on COVID-19 vaccine uptake, we used Intention-to-Treat (ITT) analyses in Ukraine and India. All oblasts and districts that were randomized to the intervention were compared to the oblasts and districts that were randomized to the control group.

In Ukraine, this meant that half of the oblasts received 5 weeks of intervention and the other half received 10 weeks of intervention. We used Poisson regression to quantify the association between the time-varying intervention and the proportion of the eligible population (i.e., those who had not yet received a first/second dose) that received the first/second dose during a given week [35, 36]. Specifically, we modeled the number of individuals who received a first/second dose of the vaccine in each week as a function of the binary intervention variable (i.e., 1: intervention week; 0: non-intervention week), weekly categorical indicator variables to account for long term trend in vaccinations across all oblasts, oblast-specific random effects to account for differences at baseline, observation-level random effects to account for overdispersion in the count data, and the eligible population (log scale) as an offset term. We considered a lagged relationship between the outcome and the intervention variable to account for the possibility that it takes time once an intervention begins to see its effect more widely. Therefore, instead of including the intervention indicator on the same week as the coverage outcome, we included the intervention status from the week before (i.e., one week lag). The model was fitted in R Statistical Software (R Core Team 2022).

In India, with the parallel RCT design, we compared the intervention group to the control group, using the same Poisson regression analysis as previously described for Ukraine. We once again used a one-week lag for the intervention variable. However, the intervention variable differed in the India analysis as some regions never received intervention (i.e.,

intervention variable was equal to zero for all weeks) while others received intervention during all weeks (i.e., intervention was equal to one for each week other than the first week due to the lag being used).

In Pakistan, with the pre-post design, we compared the uptake in during the intervention period to the period before the intervention using a Poisson regression model with random effects, 1-week lag for the intervention and an interaction term between intervention and time to determine whether the intervention caused an increase in the vaccine coverage.

**Secondary outcomes.** The survey data in Ukraine were analyzed using logistic regression to test whether the intervention impacted self-reported vaccine uptake (*'Have you received a COVID-19 vaccine?'*). Respondents could answer 'yes', 'no, but I have an appointment' and 'no'—the first 2 answers were combined and compared to 'no'.

We also analyzed the effect of the intervention on vaccination intention among those who said they were unvaccinated, using ordinal logistic regression modeling (*'How likely are you to get a COVID-19 vaccine?'*, answers on a 5-point Likert scale from 'extremely likely' to 'extremely unlikely').

The Vaccine Trust Indicator (VTI) is a validated 6-item scale measuring trust in various aspects of vaccines (see scale in the survey in the Supporting Information) [25]. On all items respondents could score between 0–10. An average score was then created and categorized into three groups: low trust (scoring less than four), medium trust (between four and seven), and high trust (seven and above). We analyzed whether there was a difference between oblasts in VTI scores using ordinal logistic regression. We also used ordinal logistic regression to determine whether there was an association between VTI scores and vaccine uptake and uptake intentions.

Trust in various key information sources about COVID-19 vaccination (the government, media, Ministry of Health, UNICEF, physician/family doctor, family/friends) was asked with the question '*How much do you trust each of the following with the information that they provide about COVID-19 vaccination?'*. Respondents could answer on a five-point Likert scale, ranging from 'none at all', to 'a great deal'. This was grouped into low trust, neutral and high trust. Logistic regression models were created to test the associations between both high trust and low trust (compared to neutral) in the stakeholders and self-reported vaccination.

All analyses were adjusted for time (i.e., survey wave), study step (no intervention, intervention), age group (18–29, 30–44, 45–59, 60+), sex (male, female) and region (West, Center, East, South). Oblast was added to the model as a fixed effect.

**Tertiary outcomes.** In all countries, data were obtained from Meta on:

- Reach: how many individuals were reached with the ads

- Impressions: how many times users were reached in total with the ads

- Clicks and clickthrough rate: clicking on link to access further information or schedule a vaccination appointment, the clickthrough rate was calculated by dividing the number of clicks by the number of impressions

Primary, secondary and tertiary analyses were registered before data were collected at OSF: https://osf.io/27t6r/.

**Sensitivity analyses/non-registered analyses.** We carried out a non-registered sensitivity analysis in Ukraine, whereby we look at actual exposure to the intervention, which was determined by reach and traffic of the ads in each oblast. Due to data availability, this analysis was limited to Ukraine.

## Results

### Descriptive overview of intervention data

In Ukraine, the ads reached 5.3 million people in the first five weeks of the intervention, and 10.7 million people in the second five weeks of the intervention. Users were reached on average 3.05 times in the first five weeks, compared to 3.23 times in the second 5-week period. In India, the ads reached 42.8 million people, and were seen an average of 13.0 times. In Pakistan, the ads reached 52.3 million users overall, and were seen on average 5.2 times in Urdu, and 4.6 times in English. A full overview of the metrics can be found in S5 Table.

### Primary results by country

In Ukraine, the COVID-19 vaccination second dose coverage at the start of the study varied from less than 10% in some oblasts to more than 35% in Kyiv. There was an increasing trend in coverage across all oblasts, however the rate of increase slowed over the course of the study, see Fig 2. The intervention did not impact the likelihood that individuals who had not previously received the first or second dose would do so during the study period (first dose Relative Risk (RR): 1.00, 95% Confidence Interval (CI) 0.92–1.08; second dose RR: 0.93, 95% CI 0.86–1.01).

In India, the baseline COVID-19 vaccination second dose coverage ranged between 20% and more than 80% in the 174 included districts, see Fig 3. During the intervention period, there was only a small increase in coverage overall: from mean 52.7% second dose coverage (standard deviation (SD) 13.1) at the start of intervention to 55.2% (SD 12.7) at end of intervention. However, this change was not attributable to the intervention (first dose RR: 0.93, 95% CI 0.75–1.17; second dose RR: 0.85, 95% CI 0.70–1.04).

In Pakistan, where we had a pre-post design, the second dose coverage ranged between less than 10% and more than 60% at baseline, see Fig 4. We similarly found that the intervention did not influence COVID-19 vaccination coverage for the second dose (RR: 0.64; 95% CI: 0.01–29.9). There was also no effect of the intervention for first dose coverage (RR: 2.1; 95% CI: 0.08–59.7).

As a sensitivity analysis in Ukraine, we repeated the primary analysis while quantifying the intervention exposure in each oblast using the reach of the campaign in each week divided by the total oblast population. Similar to the primary analysis however, there was little to no impact of the intervention on vaccinations among the eligible population (RR: 0.98, 95% CI 0.95–1.02).

### Secondary outcome: Ukraine

A total of 140,783 people responded to the survey: 56,631 for Survey 1, 43,787 for Survey 2, and 40,365 for Survey 3. The majority of respondents were female (85%) and 63% were aged between 30–59 years (mean 45 years, SD 13.8), see S6 Table. Averaged across the three surveys, 68% said they were vaccinated against COVID-19, 28% said they were not, and 4% indicated to have scheduled an appointment to get vaccinated. The share of vaccinated people increased over time (63% in Survey 1, 69% in Survey 2, and 73% in Survey 3), which was a statistically significant increase compared to Survey 1 (Survey 2: adjusted Odds Ratio (aOR) 1.25, 95% Confidence Interval (CI) 1.19–1.31, Survey 3: aOR 1.44, 95% CI 1.34–1.54), Table 1 & S6 Table.

The intervention had no impact on the self-reported COVID-19 vaccination uptake among the survey respondents (aOR: 0.98, 95% CI: 0.93–1.04), see Table 1. The share of the unvaccinated respondents who said it was unlikely that they would get vaccinated in the future

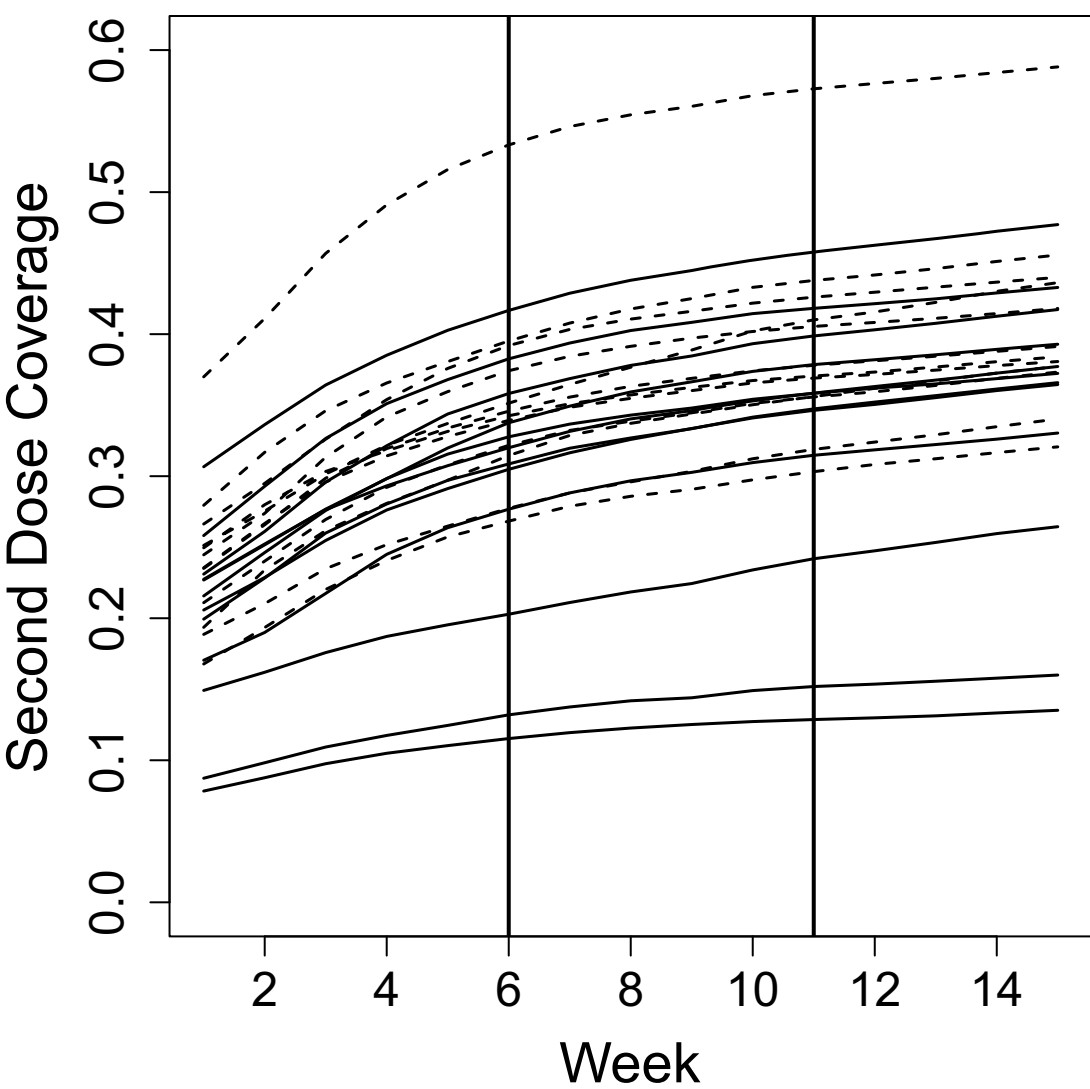

**Fig 2. Second dose COVID-19 vaccination coverage in the 24 oblasts in Ukraine during the study.** Second dose COVID-19 vaccination coverage by oblast. The dotted lines represent the oblasts that received the intervention for 10 weeks, the straight lines represent the oblasts that received the intervention for 5 weeks. The vertical lines at week 6 and week 11 mark the start of Step 1 and Step 2.

increased slightly over time. There was no effect of the intervention on the likelihood that unvaccinated respondents would get vaccinated (aOR 0.97, 95% CI 0.88–1.07), S7 Table.

The share of those scoring high on the Vaccine Trust Indicator (VTI) increased slightly over time (42% in Survey 1, 45% in Survey 2, 46% in Survey 3), but this was not due to the intervention (aOR: 0.95, 95% CI 0.90–1.01). We did find that scoring medium or high on the VTI, compared to scoring low, was strongly associated with self-reported vaccine uptake (medium aOR: 4.68, 95% CI: 4.49–4.87, high aOR: 58.94, 95% CI: 55.14–62.99), see Table 2. Similarly, VTI predicted vaccination intentions among the unvaccinated respondents (medium VTI aOR: 6.47, 95% CI: 6.06–6.91, high VTI aOR: 29.98, 95% CI: 25.98–34.67).

High trust in vaccination stakeholders was significantly associated with self-reported vaccine uptake for all information sources (e.g., UNICEF aOR: 5.09, 95% CI: 4.71–5.51), see S8 Table. The inverse was also true; those with low trust in information stakeholders were less

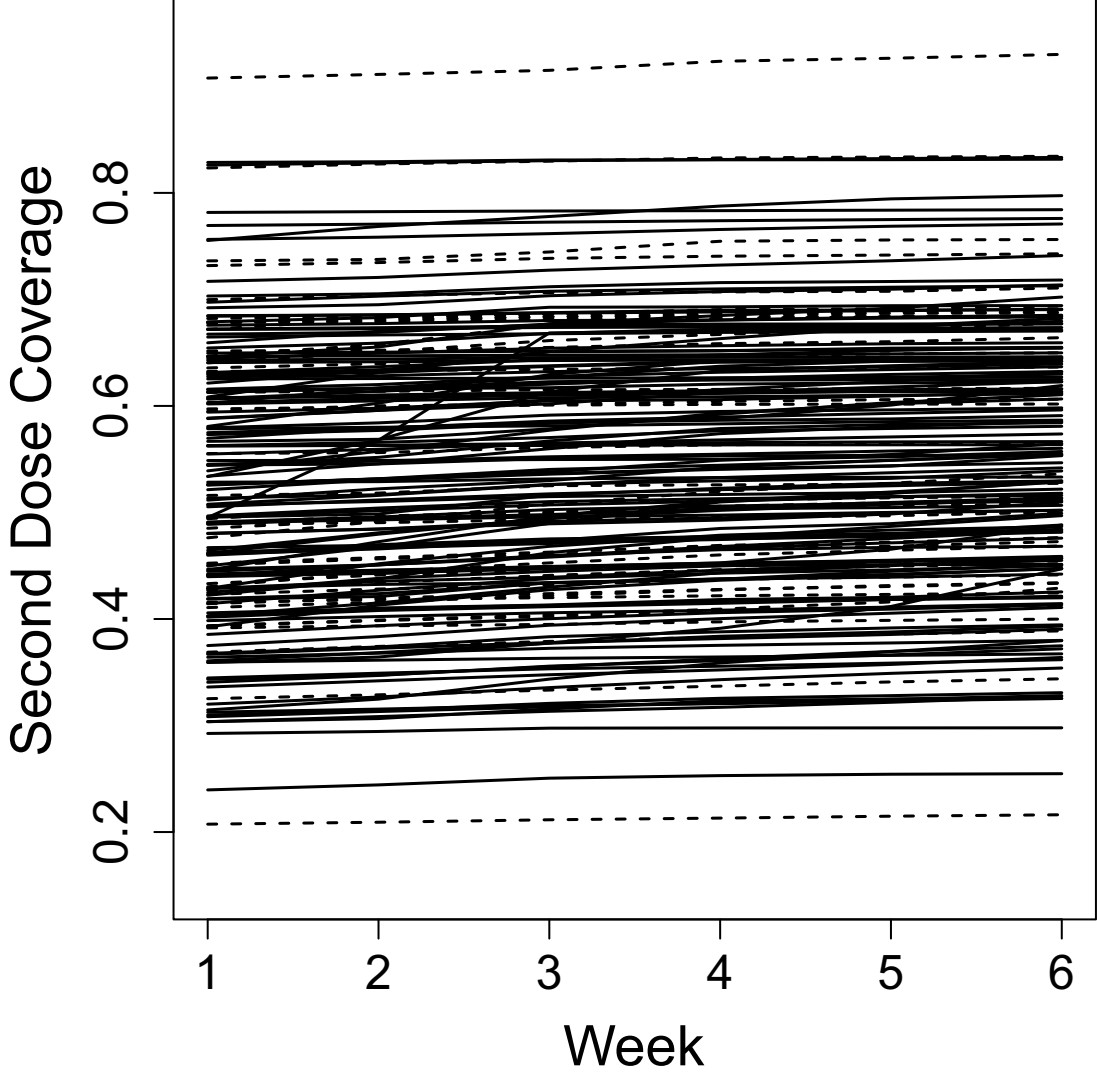

**Fig 3. Second dose COVID-19 vaccination coverage in the 174 districts in India during the study.** Second dose COVID-19 vaccination coverage by district during the 6-week intervention period. The dotted lines represent the districts that received the intervention, the straight lines were control districts.

likely to be vaccinated against COVID-19 (e.g., government aOR: 0.21,95% CI 0.20–0.22), see S9 Table.

## Discussion

In three large studies, of which two were RCTs, we found that an intervention comprising persuasive posts promoting COVID-19 vaccination on Facebook, delivered by a trusted source in the form of UNICEF, did not influence COVID-19 vaccination coverage in Ukraine, India, and Pakistan. While disappointing, the results are consistent with other, smaller studies using social media interventions for behavior change [15, 20, 37, 38].

A possible explanation for the observed null results is that ads on Facebook alone are an insufficient vehicle to influence actual immunization coverage. Facebook usage varies strongly across countries, and in India and Pakistan it is dominated by men. Furthermore, while the

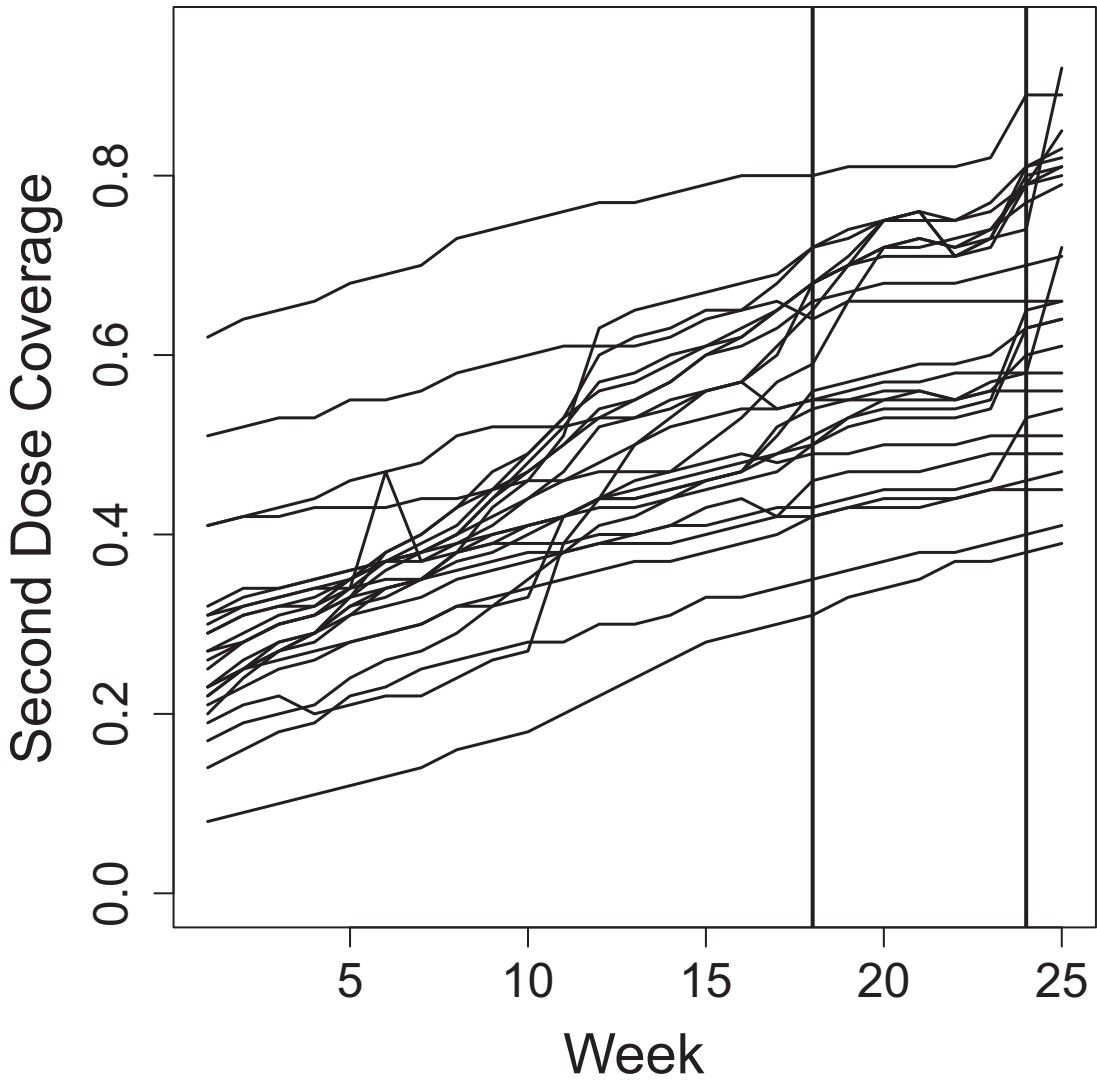

**Fig 4. Second dose COVID-19 vaccination coverage in the 25 city districts in Pakistan during the study.** Second dose COVID-19 vaccination coverage by city district during the study period. The vertical lines at week 18 and 24 mark the start and the end of the intervention period.

metrics indicated that the ads were seen multiple times, this might not have been enough to create a lasting impression. Vaccine decision-making is inherently complex, and the sole influence of persuasive messaging delivered via online ads may not have been enough to affect behavior [38].

The timing of the interventions may also have limited the effect of the interventions. Given the timing of our study in early 2022, many people who were eager to get vaccinated may have already done so, and those who initially opted to wait were likely to also have been vaccinated. This implies that the target audience may have shifted and been composed of people who were less intentioned to get vaccinated–a more difficult audience to convince. On the other hand, the timing of the intervention may have meant that there was to a large extent no unmet demand, which would imply that any changes we observed in vaccination coverage would be linked to the intervention.

**Table 1. Associations between the intervention and self-reported COVID-19 vaccine uptake in Ukraine.**

| | Odds Ratio 95% CI | P-value | Adjusted* Odds Ratio 95% CI | P-value |
|---|---|---|---|---|
| **Randomized group** | | | | |
| 5-weeks | Reference | | Reference | |
| 10-weeks | 1.31 (1.27–1.35) | 0.000 | 0.98 (0.93–1.04) | 0.573 |
| **Survey time** | | | | |
| Survey 1 | Reference | | Reference | |
| Survey 2 | 1.23 (1.19–1.28) | 0.000 | 1.25 (1.19–1.31) | 0.000 |
| Survey 3 | 1.43 (1.38–1.48) | 0.000 | 1.44 (1.34–1.54) | 0.000 |
| **Region** | | | | |
| West | Reference | | Reference | |
| Central | 1.01 (0.92–1.11) | 0.863 | 1.00 (0.92–1.09) | 0.958 |
| East | 0.96 (0.85–1.09) | 0.517 | 0.95 (0.84–1.07) | 0.364 |
| South | 0.89 (0.80–0.98) | 0.023 | 0.87 (0.79–0.96) | 0.006 |
| **Age** | | | | |
| 18–29 | Reference | | Reference | |
| 30–44 | 0.93 (0.89–0.97) | 0.001 | 0.94 (0.91–0.99) | 0.010 |
| 45–59 | 0.96 (0.92–1.00) | 0.048 | 0.98 (0.94–1.02) | 0.309 |
| 60+ | 0.80 (0.76–0.84) | 0.000 | 0.82 (0.79–0.87) | 0.000 |
| **Gender** | | | | |
| Male | Reference | | Reference | |
| Female | 1.47 (1.42–1.52) | 0.000 | 1.45 (1.40–1.51) | 0.000 |

CI = Confidence Interval

*Adjusted for: all other variables reported in the table and oblast

Our persuasive messages were selected based on extensive testing using Facebook's Brand Lift Studies, which showed positive attitudinal changes towards COVID-19 vaccines after exposure to the selected ads (manuscript in submission). This is in line with a recent review of 819 randomized experiments of COVID-19 messaging campaigns from public health agencies on Facebook and Instagram, finding that these campaigns were effectively influencing self-reported belief [39]. However, it seems that attitudinal changes in this instance were not sufficient for actual vaccine uptake, which may be another demonstration of the attitude-behavior gap [40, 41]. It can also be that both the timing of the actual interventions (months after the testing though BLS), as well as the changed target population of unvaccinated adults have

**Table 2. Association between Vaccine Trust Indicator and self-reported COVID-19 vaccination status and intention to receive a COVID-19 vaccine among the unvaccinated in Ukraine.**

| | Vaccination status | | Vaccination intention | |
|---|---|---|---|---|
| | Adjusted* OR (95% CI) | P-value | Adjusted* OR (95% CI) | P-value |
| **Vaccine Trust Indicator** | | | | |
| Low | Reference | - | Reference | - |
| Medium | 4.68 (4.49–4,87) | <0.001 | 6.47 (6.06–6.91) | <0.001 |
| High | 59.94 (55-14-62.99) | <0.001 | 29.98 (25.92–34.67) | <0.001 |

OR = Odds Ratio, CI = Confidence Interval

Vaccination status: logistic regression. Vaccination intention: ordinal logistic regression

*adjusted for: randomized group, time (i.e., survey wave), region, age, gender, oblast

rendered the ads less powerful. A similar mechanism happened in a study testing persuasive messages among vaccine hesitant white evangelicals in the United States, which showed initial success using messages that emphasized community and reciprocity [42]. However, the effect disappeared once COVID-19 vaccines were introduced–likely due to more extreme views towards vaccines among a more hesitant unvaccinated group [42]. Context and timing of persuasive messaging requires precision.

Furthermore, external events may have influenced COVID-19 vaccine uptake, including the looming threat of a Russian invasion in Ukraine and the large Omicron wave in all three countries that coincided with our intervention [43, 44]. The Omicron wave may have spurred the general public (i.e., in both intervention and control districts and oblasts) to get vaccinated, which may have biased an effect to the null. Behavior is inherently complex and is influenced by various factors, from internal considerations to socio-economic circumstances and previously described external factors [29]. Interventions such as ours likely play a small role in the decision-making process of vaccination practices.

A recent systematic review of the effect of social media interventions on vaccine behavior found that interventions that were based on educational messages were not effective in triggering changes in vaccine behavior [38]. On the other hand, social media interventions that were successful in eliciting behavior change used multi-pronged approaches, combining for instance online messaging with offline campaigns [38]. Future studies should test multi-pronged and cross-platform interventions, combining complementary on- and offline campaigns, as well as more geo-targeted campaigns that will resonate on a more local level [45, 46].

Furthermore, customizing and tailoring messages to specific audiences (e.g., more hesitant adults) could help to increase the effectiveness of the intervention, although effect sizes of such customization might be small [47, 48]. A recent RCT testing messages that were customized to participants' values and vaccine attitudes did not increase childhood immunization rates [37]. It even found a slight negative effect among more vaccine hesitant parents, which was interpreted as a possible overreach of the customized messages, rendering them less effective [37]. Customization should therefore be carried out with care and requires a thorough understanding of the target audiences.

The analyses based on the survey in Ukraine showed the importance of trust in vaccines and in various vaccination stakeholders; those expressing higher trust were more likely to be vaccinated or, among the unvaccinated, intended to get vaccinated. This is in line with previous research, showing that trust is crucial for public health and vaccine acceptance [49–54]. As evidence around effective interventions to increase or leverage trust for vaccination behavior is to date lacking [55], the individual indicators of the Vaccine Trust Indicator short scale (e.g., 'vaccination forms part of a healthy lifestyle') could point to potential levers. Our studies further validated the VTI and showed its utility in providing relevant measures of vaccine trust in populations [25].

## Strengths & limitations

The main strengths of our studies include the fact that we were able to use two randomized controlled trials to robustly evaluate the effect of our persuasive messaging campaigns on Facebook. To our knowledge, this is the first set of studies that have investigated the direct effects of social media interventions on a national level in three different LMICs. The studies naturally have several limitations. Firstly, we randomized on oblasts and districts level in Ukraine and India, but the intervention may have spilled over to control oblasts and districts as the geographical targeting of the intervention cannot so precisely be carried out in Facebook's Ads Manager. The quality of the COVID-19 vaccination data in Pakistan was suboptimal with

decreases in cumulative vaccination coverage noted at certain weeks for some districts reflecting data entry errors, which may have hampered our ability to detect an effect. While two of the studies were RCTs, they are among the first in this field to use this type of study design. More RCTs in this field are needed to further establish causality. We found that survey recruitment through ads on Facebook resulted in a non-representative sample. While 68% of respondents in Ukraine said they were vaccinated against COVID-19, national statistics point to a 30% vaccine coverage around that time [56]. Similarly, 85% of the respondents in Ukraine identified as women. Suggested methods in the literature to increase the representativeness of surveys recruited through Facebook include stratification of ads campaigns and selectively (de-)activating ad campaigns during the recruitment phase [57].

## Conclusion

Three nationwide studies using pre-tested, persuasive pro-COVID-19 vaccine messaging on Facebook did not result in an increase in regional COVID-19 vaccination coverage in Ukraine, India, and Pakistan. Our campaigns reached more than 110 million Facebook users and garnered 2.9 million clicks, signaling the broad potential reach of digital ad campaigns, but the results underscore that reach alone is not enough to shift behaviors. Gains might be made by using multi-pronged demand generation strategies, which include more customized messaging. Trust in vaccines and in sources of vaccination information was an important predictor of vaccination practices and should be leveraged in future vaccination campaigns.

## Supporting information

**S1 Table. Ukrainian oblasts included in the study.**
(DOCX)

**S2 Table. Baseline COVID-19 vaccination coverage in Ukraine.**
(DOCX)

**S3 Table. Selected intervention districts five state trial through GeoLift in India.**
(DOCX)

**S4 Table. Districts in Pakistan included in the study.**
(DOCX)

**S5 Table. Ad Metrics in Ukraine, India, and Pakistan.**
(DOCX)

**S6 Table. Demographics of survey in Ukraine.**
(DOCX)

**S7 Table. Associations between the intervention and intentions to receive a COVID-19 vaccine among the unvaccinated in Ukraine.**
(DOCX)

**S8 Table. Association between high trust in stakeholders and self-reported vaccination status in Ukraine.**
(DOCX)

**S9 Table. Association between low trust in stakeholders and self-reported vaccination status in Ukraine.**
(DOCX)

**S1 Fig. Ukraine study design.**
(TIFF)

**S1 Text. Questionnaire for Ukraine.**
(DOCX)

**S2 Text. Inclusivity in global research questionnaire.**
(DOCX)

## Author Contributions

**Conceptualization:** Sarah Christie, Chelsey Lepage, Amyn A. Malik, Scott Bokemper, Surangani Abeyesekera, Brian Boye, Midhat Moini, Zara Jamil, Taha Tariq, Tamara Beresh, Ganna Kazymyrova, Liudmyla Palamar, Alexandra Faller, Andreea Seusan, Erika Bonnevie, Joe Smyser, Kadeem Khan, Mohamed Gulaid, Sarah Francis, Angus Thomson, Saad B. Omer.

**Data curation:** Sarah Christie, Amyn A. Malik, Scott Bokemper, Brian Boye, Midhat Moini, Zara Jamil, Taha Tariq, Tamara Beresh, Ganna Kazymyrova, Liudmyla Palamar, Elliott Paintsil, Alexandra Faller, Andreea Seusan, Erika Bonnevie, Joe Smyser.

**Formal analysis:** Maike Winters, Amyn A. Malik, Joshua L. Warren.

**Funding acquisition:** Chelsey Lepage.

**Investigation:** Maike Winters, Amyn A. Malik, Scott Bokemper.

**Methodology:** Sarah Christie, Scott Bokemper, Surangani Abeyesekera, Joe Smyser, Kadeem Khan, Mohamed Gulaid, Sarah Francis, Angus Thomson, Saad B. Omer.

**Project administration:** Sarah Christie.

**Supervision:** Sarah Christie, Saad B. Omer.

**Writing – original draft:** Maike Winters, Sarah Christie, Joshua L. Warren, Saad B. Omer.

**Writing – review & editing:** Chelsey Lepage, Amyn A. Malik, Scott Bokemper, Surangani Abeyesekera, Brian Boye, Midhat Moini, Zara Jamil, Taha Tariq, Tamara Beresh, Ganna Kazymyrova, Liudmyla Palamar, Elliott Paintsil, Alexandra Faller, Andreea Seusan, Erika Bonnevie, Joe Smyser, Kadeem Khan, Mohamed Gulaid, Sarah Francis, Angus Thomson.

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
