## [Decision Letter · Decision Letter 0]

2 May 2023

PGPH-D-23-00277

Persuasive COVID-19 vaccination campaigns on Facebook and nationwide vaccination coverage in Ukraine, India, and Pakistan

Dear Dr. Winters,

Thank you for submitting your manuscript to PLOS Global Public Health. After careful consideration, we feel that it has merit but does not fully meet PLOS Global Public Health’s publication criteria as it currently stands. Therefore, we invite you to submit a revised version of the manuscript that addresses the points raised during the review process.

We look forward to receiving your revised manuscript.

Kind regards,

Karen A. Grépin, Ph.D.

Academic Editor

Journal Requirements:

1. Please include a complete copy of PLOS’ questionnaire on inclusivity in global research in your revised manuscript. Our policy for research in this area aims to improve transparency in the reporting of research performed outside of researchers’ own country or community. The policy applies to researchers who have travelled to a different country to conduct research, research with Indigenous populations or their lands, and research on cultural artefacts. The questionnaire can also be requested at the journal’s discretion for any other submissions, even if these conditions are not met.  Please find more information on the policy and a link to download a blank copy of the questionnaire here: https://journals.plos.org/plosone/s/best-practices-in-research-reporting. Please upload a completed version of your questionnaire as Supporting Information when you resubmit your manuscript.

2. Please send a completed 'Competing Interests' statement, including any COIs declared by your co-authors. If you have no competing interests to declare, please state "The authors have declared that no competing interests exist". Otherwise please declare all competing interests beginning with the statement "I have read the journal's policy and the authors of this manuscript have the following competing interests:"

3. Please provide separate figure files in .tif or .eps format.

Additional Editor Comments (if provided):

Dear Authors,

Thank you for submitting this article for consideration at PLOS Global Public Health. As you can see from the reviewer comments, both reviewers felt this was a strong paper but had what I felt were very useful suggestions to improve the framing and presentation of the article. I agree with all these comments. I have also attached some of my own feedback below. As a result, I am recommending a minor revision of the text. If the edits are made accordingly, I believe there is a good chance this article could be accepted at this journal.

Karen Grépin

Comments from Editor:

• How did you account for time in your randomized treatment? Those who received 10 weeks of treatment also had 5 more weeks of time to get vaccinated.

• In Ukraine, what was the timing of the 5 weeks of intervention in relation to the 10-week intervention? Was it at the start/finish?

• Ideally, you would have also had a control group that received no treatment to account for the additional time without any intervention. I think it needs to be made clearer that your intervention allows you to test the impact of 5 additional weeks of exposure to the intervention, not necessarily of the intervention itself.

• Is it possible to provide balance tables to demonstrate how well the randomization was implemented? There is a risk due to the small number of geographic units that it was not.

• I struggled to interpret table 2. Could these findings be presented in a different way?

• Why include Pakistan in this paper at all? The data are not randomized and really should be treated separately. Its outcomes are not even presented in the main tables. It feels like a vestige that might need dissection.

• Why not also test whether the intervention affected vaccine intention using the randomization? This seems like missed opportunity.

• I did not like this sentence: “This means that most people who 237 were eager to get vaccinated had already done so and those who initially opted to wait were likely also 238 vaccinated. This implies that the target audience may have been composed of people who were more 239 hesitant and less intentioned to get vaccinated – a more difficult audience to convince.” But in theory, that is exactly the audience the intervention aimed to convince, no?

Reviewers' comments:

Reviewer's Responses to Questions

**Comments to the Author**

1. Does this manuscript meet PLOS Global Public Health’s publication criteria? Is the manuscript technically sound, and do the data support the conclusions? The manuscript must describe methodologically and ethically rigorous research with conclusions that are appropriately drawn based on the data presented.

Reviewer #1: Yes

Reviewer #2: Yes

2. Has the statistical analysis been performed appropriately and rigorously?

Reviewer #1: Yes

Reviewer #2: Yes

3. Have the authors made all data underlying the findings in their manuscript fully available (please refer to the Data Availability Statement at the start of the manuscript PDF file)?

Reviewer #1: Yes

Reviewer #2: Yes

4. Is the manuscript presented in an intelligible fashion and written in standard English?

Reviewer #1: Yes

Reviewer #2: Yes

5. Review Comments to the Author

Reviewer #1: Summary:

This study has tried to assess, whether persuasive messages from UNICEF, disseminated via Facebook has any impact on the Covid 19 vaccine uptake in Ukraine, India and Pakistan. The study was designed as an RCT in Ukraine and India and in Pakistan it was deployed as a pre-post design with varying intervention periods in the three countries. The study concluded that there was no effect of intervention on vaccination uptake in all the three countries

General Remarks:

A well written piece of work. Authors have explored an interesting and important research area, which is very suitable in the current era, where internet and social media rules the world.

Comments:

Methods section to be included after introduction and objectives and before results

A small introduction about the study setting (especially about social media usage in the three countries) will help the international readers in understanding the social media use/influence in these countries

Explain, why the intervention period and the study design are differing for the countries included in the study

Specify, how informed consent was obtained from the participants for the online survey

Reviewer #2: I have attached a file containing all of my comments to the authors.

Overall, I appreciate the endeavor on a large scale study and am eager to have it be added to the infodemic space. I have general concerns about the framing and scope of the study, and minor suggestions on improving the methods.

6. PLOS authors have the option to publish the peer review history of their article (what does this mean?). If published, this will include your full peer review and any attached files.

**Do you want your identity to be public for this peer review?** For information about this choice, including consent withdrawal, please see our Privacy Policy.

Reviewer #1: No

Reviewer #2: No

---

## [Decision Letter · Decision Letter 1]

15 Aug 2023

Persuasive COVID-19 vaccination campaigns on Facebook and nationwide vaccination coverage in Ukraine, India, and Pakistan

PGPH-D-23-00277R1

Dear Dr Winters,

We are pleased to inform you that your manuscript 'Persuasive COVID-19 vaccination campaigns on Facebook and nationwide vaccination coverage in Ukraine, India, and Pakistan' has been provisionally accepted for publication in PLOS Global Public Health.

Best regards,

Abram L. Wagner, PhD, MPH

Academic Editor

Editor comments:

Although not required, you may want to consider putting dates into the abstract.

Reviewer Comments (if any, and for reference):

Reviewer's Responses to Questions

**Comments to the Author**

1. If the authors have adequately addressed your comments raised in a previous round of review and you feel that this manuscript is now acceptable for publication, you may indicate that here to bypass the “Comments to the Author” section, enter your conflict of interest statement in the “Confidential to Editor” section, and submit your "Accept" recommendation.

Reviewer #2: All comments have been addressed

2. Does this manuscript meet PLOS Global Public Health’s publication criteria? Is the manuscript technically sound, and do the data support the conclusions? The manuscript must describe methodologically and ethically rigorous research with conclusions that are appropriately drawn based on the data presented.

Reviewer #2: Yes

3. Has the statistical analysis been performed appropriately and rigorously?

Reviewer #2: Yes

4. Have the authors made all data underlying the findings in their manuscript fully available (please refer to the Data Availability Statement at the start of the manuscript PDF file)?

Reviewer #2: No

5. Is the manuscript presented in an intelligible fashion and written in standard English?

Reviewer #2: Yes

6. Review Comments to the Author

Reviewer #2: The authors have addressed all my comments and the piece is now ready for publication.

7. PLOS authors have the option to publish the peer review history of their article (what does this mean?). If published, this will include your full peer review and any attached files.

**Do you want your identity to be public for this peer review?** For information about this choice, including consent withdrawal, please see our Privacy Policy.

Reviewer #2: No
